# Evaluating Knowledge of Human Microbiota among University Students in Jordan, an Online Cross-Sectional Survey

**DOI:** 10.3390/ijerph182413324

**Published:** 2021-12-17

**Authors:** Anas H. A. Abu-Humaidan, Jawad A. Alrawabdeh, Laith S. Theeb, Yazan I. Hamadneh, Mohammad B. Omari

**Affiliations:** 1Department of Pathology, Microbiology, and Forensic Medicine, School of Medicine, The University of Jordan, Amman 11942, Jordan; 2School of Medicine, The University of Jordan, Amman 11942, Jordan; joa0194878@ju.edu.jo (J.A.A.); lyt0191698@ju.edu.jo (L.S.T.); yzn0190785@ju.edu.jo (Y.I.H.); mhm0193400@ju.edu.jo (M.B.O.)

**Keywords:** microbiota, knowledge assessment, university students, microbiology literacy

## Abstract

Human microbiota have a significant impact on the health of individuals, and reciprocally, lifestyle choices of individuals have an important effect on the diversity and composition of microbiota. Studies assessing microbiota knowledge among the public are lacking, although it is hypothesized that this knowledge can motivate healthier behavior. Hence, this study aimed to measure microbiota knowledge among university students, and the effect of this knowledge on behavioral beliefs. A descriptive cross-sectional study was conducted among students from various fields of study enrolled at the University of Jordan, using an online questionnaire. The questionnaire consisted of 3 parts: demographics, general knowledge of microbiota, and behavioral beliefs related to microbiota. Four hundred and two responses were collected from verified university students. Participants were divided into two groups depending on whether they took a formal microbiology course (45 h) or not. Results from those two groups were compared using appropriate statistical methods. Results showed that most participants, even those who did not take a formal microbiology course, displayed good knowledge of what microbiota is and how they can be influenced by personal and environmental factors. Participants who took a microbiology course had significantly higher microbiota knowledge scores and were more aware of the effect of antibiotics on microbiota. Participants’ behavioral beliefs regarding their antibiotic use, but not their diet and lifestyle choices, were affected by their knowledge of microbiota. The study indicates that disseminating knowledge regarding microbiota and microbiology in general, can improve behaviors related to antibiotic use.

## 1. Introduction

Microorganisms have a great impact on human health and disease, whether as part of the microbiota on the skin, gut, and mucosal surfaces of humans, or as free-living infectious and non-infectious agents. Discovery of the role of microorganisms in transmitting infections in the late 19th century caused a gradual shift in public attitudes regarding infectious disease control and pathogenesis [1], which consequently led to a significant reduction in rates of transmissible diseases [2]. Similarly, the past few decades have witnessed great steps in the understanding of human microbiota, aided by molecular and biochemical tools that elucidated the diversity and variation of microbes across populations and health states [3]. The wealth of data available indicates that everyday choices regarding diet, exercise, and hygiene can affect human microbiota [4,5]. Indeed, individuals have a unique microbial signature that depends largely on their interactions with the environment [6,7]. 

Perturbation of microbiota was shown to be associated with several diseases, such as inflammatory bowel diseases [8], atopic dermatitis [9], and cancer [10] among others. Knowledge of microbiota and how they affect health can lead individuals to healthier lifestyle choices, such as consuming a high fiber diet [11], exercising regularly [12], or using antibiotics wisely [13]. This is supported by studies measuring knowledge in health-related issues such as antibiotic use or precautionary measures towards infectious diseases, which indicate that knowledge of a certain issue is often manifested in behavior and attitudes [14,15]. Knowledge of microbiological concepts among the public, including microbiota, is expected to affect decisions on a social level as well, since it can guide decision-makers [16].

Previous studies related to knowledge of microbiota focused mainly on the therapeutic aspect of microbiota, such as probiotics [17,18], and fecal microbiota transplant (FMT) [19,20], with the populations tested consisting mainly of professionals and students in healthcare. 

This descriptive cross-sectional study aimed to measure knowledge of university students from various fields of study on the topic of microbiota, firstly, to understand how students perceived microbiota and how they are affected by antibiotics, and secondly, to assess students’ attitudes towards consuming probiotics, as well as behavioral beliefs regarding the effect of microbiota knowledge on lifestyle choices, diet, and antibiotic use.

## 2. Methods

### 2.1. Study Population and Design 

This was a descriptive cross-sectional study. The sample consisted of students from all schools enrolled at the University of Jordan. The University of Jordan has around 49,000 students enrolled in various programs across 24 schools. Data were collected from healthcare schools (e.g., Medicine, Dentistry, Pharmacy, Nursing), and non-healthcare schools (e.g., Physics, Mechanical engineering, Law, Business, Arts). 

The study was conducted using an online questionnaire created on Microsoft Forms and delivered to students in the period between 31 January and 9 February. In order to ensure that participants are currently students at the University of Jordan, they needed to sign in with their university e-mail that only contains their initials and their student ID, which cannot be used to identify the students. The questionnaire was posted on all online groups that the students use for communication within each school. In addition, representatives from each school were contacted to help us distribute the questionnaire among students. 

### 2.2. Questionnaire Design 

The first page of the questionnaire included an informed consent form, as well as a description of the contents of the questionnaire, what it aims to study, and the confidentiality of the data. It was written in both Arabic and English. The questionnaire was at first pilot tested on 30 students, and modifications were done accordingly. The questionnaire consisted of 33 questions divided into three sections: (1) Demographics and general information about the participant, (2) general knowledge about bacteria, microbiota, and antibiotics, and (3) general attitude and lifestyle choices that can affect microbiota. 

The demographics and general information section included questions about age, sex, field of study, academic year, and the level of knowledge about bacteria. The knowledge level choices were (Poor knowledge, if you never read or taken a course that discusses bacteria. Basic knowledge, from personal reading or biology classes. Advanced knowledge, if you took at least one microbiology course at university). The questionnaire also studied if the participants would like to learn more about microbiota, and what sources they would prefer to learn from.

The knowledge section consisted of 15 questions with 5 possible answers for each question: “I don’t know, definitely not, probably not, probably, and definitely”. The answers “probably and definitely” are referred to as “agree” in the results section and the answers “definitely not and probably not” are referred to as “disagree”, while the answer “I don’t know” is referred to as “uncertain”. The statements used estimated the participants’ knowledge about microbiota as well as the effect of antibiotics on microbiota. To score microbiota knowledge, correct and incorrect answers were scored as 1 and 0, respectively.

The attitude section consisted of 10 questions with 4 answers for each question “definitely not, probably not, probably, and definitely*”*. The answers “probably and definitely” are referred to as “agree” in the results section and the answers “definitely not and probably not” are referred to as “disagree”. The attitude section contained statements to assess participants’ attitudes towards consuming probiotics, as well as behavioral beliefs regarding the effect of microbiota knowledge on lifestyle choices, diet, and antibiotic use. The questionnaire can be found in the Appendix A

### 2.3. Ethical Approval

This study was approved by the Institutional Review Board (IRB) at the University of Jordan (Ref. No. 2021-86). An informed consent and an explanation of voluntary participation in the study were presented at the start of the questionnaire. Collected data were treated with confidentiality. 

### 2.4. Data Analysis 

Data generated were organized in Microsoft Excel, and statistical analysis was carried out using IBM Statistical Package for the Social Sciences (SPSS) for Windows version 25.0 (Armonk, NY, USA) and GraphPad Prism 8 (San Diego, CA, USA). 

Categorical variables were presented as count and percentages in the tables and as percentages in the text, with odds ratios and confidence intervals (95% CI) when applicable, while continuous variables were presented as (Mean ± SD). Independent sample T-tests, one-way ANOVA, and Chi-square test were used to compare knowledge and attitudes between different school categories, genders, levels of knowledge about microbiology. A *p*-value of 0.05 was adopted as a threshold for significance.

## 3. Results 

### 3.1. Demographics and General Characteristics of Participants

Four hundred and two (402) participants filled the questionnaire. Among them, 38.6% were males, whilst 61.4% were females, and participants were distributed in the age groups 17–20 years (87.1%), 21–24 years (12.7%) and one response of the age 29 (0.2%) (Table 1). Respondents were grouped by their field of study as health-related specialties (70.6%) and non-health-related (29.4%). The year of study with the greatest prevalence was the second year (51.2%). As for the self-reported knowledge level (described in methods), 39.1% of the sample reported having advanced knowledge about microbiology, while 50.2% reported they had basic knowledge, and 10.7% had poor knowledge. The distributions of gender, age groups, and years of study are comparable to university population distribution. 

### 3.2. General Knowledge of Microbiota 

First, an assessment of whether participants think of bacteria primarily as a cause of disease or not was done. To that end, the agreement with two statements “the majority of bacteria in the world do not cause disease to humans” and “bacterial cells outnumber human cells in the human body” was measured. The majority (80.6% and 50.5%, respectively) of participants agreed with the above-mentioned statements (Table 2). This indicated that most participants knew that bacteria are not always associated with disease. To expand on this point, the following statements examined if participants thought that the presence of bacteria in different locations of the human body is always a cause for disease, using statements such as “presence of bacteria on the skin will always cause disease in humans” to which around only 14.4% incorrectly agreed. Similar results were obtained when asked about the presence of bacteria in the gut (Table 2). A statement about the presence of bacteria in the brain was also added, which unlike the skin or gut is not considered to harbor microbiota. The majority of participants thought correctly that the presence of bacteria in the brain is always associated with disease (68.9%) while 24.6% were uncertain (Table 2).

The participants’ knowledge regarding the presence of bacteria in food was also assessed. Most participants (70.9%) disagreed with the statement “Healthy food should never contain any type of bacteria”, indicating that the majority of participants knew that healthy food can sometimes contain bacteria (Table 2). 

Most participants (79.6%) agreed that exercise can positively affect microbiota, indicating that participants believed that certain behaviors could influence microbiota. Moreover, 66.2% of participants agreed that differences in countries and ethnicities can affect the type of bacteria living in the human body, which showed that participants are aware to some degree of the effect of environmental and genetic factors on microbiota.

### 3.3. Knowledge of the Relationship between Antibiotics and Microbiota 

Participants’ knowledge on the relationship between antibiotics and microbiota was evaluated (statements 12–15 in Table 2). The majority of participants correctly identified the effect of antibiotics on microbiota since 91.3% agreed that antibiotics can kill beneficial bacteria, and that antibiotic use can cause disease by killing beneficial bacteria (85.1%), while a minority (20.4%) of participants thought that antibiotics can kill harmful bacteria only (Table 2). Compared to knowledge regarding antibiotics, participants knew less about the possibility of using probiotics alongside antibiotic therapy, where 23.9% of participants disagreed that bacteria can be given orally to replace beneficial bacteria killed after antibiotic therapy, and 26.9% were uncertain (Table 2), indicating the highest level of uncertainty among all statements. 

### 3.4. Knowledge Scores among Various Groups of Participants

To identify factors associated with better microbiota knowledge, the total knowledge score in various groups of participants was analyzed. This was done by scoring the 15 statements found in Table 2, where correct and incorrect answers were scored as 1 and 0, respectively. The resultant knowledge score was averaged for each group (Table 3). 

Regarding reported knowledge levels, no significant difference in knowledge score was found between basic and poor knowledge (*p* = 0.153), so they were merged into one group for analysis purposes with a score of 10.1 ± 2.5. The total knowledge score of participants who reported advanced knowledge of microbiology, representing students acquainted with the concept of microbiota through a microbiology course, was significantly higher than participants who reported basic and poor knowledge in microbiology, who represent the average university student (12.8 ± 2.0 vs. 10.1 ± 2.5, respectively, *p* < 0.001). Similarly, a comparison of participants according to their field of study was done. Students in healthcare-related fields had significantly higher knowledge scores compared to other students (11.8 ± 2.4 vs. 9.6 ± 2.6, respectively, *p* < 0.001), while no significant differences in knowledge scores among males and females were found (11.3 ± 2.7 vs. 11.1 ± 2.6, respectively, *p* = 0.438). It should be noted that most students from healthcare-related fields reported advanced knowledge (60.3%), whereas only 8.48% of students from non-healthcare related fields reported advanced knowledge (*p*< 0.001).

To identify specific knowledge gaps related to antibiotic use and its effect on microbiota, responses to statements related to antibiotic effect on microbiota were examined in different knowledge groups (Table 4). Advanced knowledge students scored 0.8 points higher than basic/poor knowledge students (3.5 vs. 2.7, respectively, *p* < 0.001). In general, advanced knowledge students were more aware of the side effects of antibiotics on microbiota than other students, since 98.1% vs. 86.9%, respectively (*p* < 0.001), agreed that antibiotics can kill beneficial bacteria, and 96.8% vs. 77.6%, respectively (*p* < 0.001), agreed that antibiotic use may cause disease by killing beneficial bacteria. Similarly, 8.3% of advanced knowledge students agreed to the statement “antibiotics only kill harmful bacteria” vs. 36.3% of basic/poor knowledge students (*p* < 0.001) (Table 4). 

### 3.5. Effect of Microbiota Knowledge on Probiotic Use and Behavioral Beliefs 

To evaluate if knowledge of microbiota can affect probiotic use, the agreement with the statement “I would ingest a pill that contains bacteria as a treatment for disease if available” was assessed in each knowledge group. Knowledge of microbiota did not seem to affect willingness to take probiotics, as there was no significant difference between advanced knowledge and basic/poor knowledge students (84.1% vs. 90.6%, respectively, *p*-value = 0.058). In addition, knowledge of microbiota did not seem to affect behavioral beliefs, as no significant difference was found between advanced knowledge and basic/poor knowledge groups in the agreement with the statement “My lifestyle choices are affected by my knowledge of beneficial bacteria in the human body” (82.2% vs. 77.1%, respectively, *p*-value = 0.207). However, more students with better knowledge agreed with the statement “My use of antibiotics is affected by my knowledge of beneficial bacteria in the human body” (96.8% vs. 85.7%, *p*-value < 0.001) (Table 5). On the other hand, no significant difference was found when respondents were asked about the effect of microbiota knowledge on their diet (68.8% vs. 66.1%, *p*-value = 0.588) (Table 5). 

### 3.6. Willingness to Learn about Microbiota 

At the end of the questionnaire, participants were asked which source they would prefer to use when learning about microbiota, with 4 possible choices: healthcare workers, news outlets, social media, and trusted medical sources such as medical journals. The majority of participants (93.2%) wanted to learn more about microbiota (Table 6). The top two sources chosen were social media (45.5%) and trusted sources (70.6%). Response percentages were similar between advanced and basic/poor knowledge students.

## 4. Discussion

Awareness of how microbes in general and microbiota specifically can affect human health is important for individuals and societies alike, since this knowledge can influence lifestyle choices of individuals and guide decision-makers in forming their policies [16,21]. In this descriptive cross-sectional study, knowledge of microbiota among university students was assessed. 

Because we did not find previous studies that assessed knowledge of microbiota in individuals outside of the healthcare field, statements that are understandable by participants who have basic knowledge in microbiology had to be generated, which proved to be challenging. For example, we had to use the word “always” in the statement “Presence of bacteria on the skin always causes disease”, because without the word “always”, answering agree or disagree can both be considered correct. Still, the questionnaire performed well in assessing microbiota knowledge, since participants who identified themselves as having advanced knowledge had significantly higher knowledge scores compared to those with basic and poor knowledge (12.8 ± 1.95 vs. 10.1 ± 2.473, respectively, *p* < 0.001). The criterion for choosing advanced knowledge in the questionnaire was attending a microbiology course at university. Therefore, it was not surprising that students who reported advanced knowledge were mostly from healthcare-related fields, since they usually attend a microbiology course during their second or third year of study. Microbiology courses at university are 45 h, and usually present the topic of microbiota and its role in health and disease. 

The study indicated that most students knew that the presence of bacteria on the skin or in the gut is not always associated with disease. Moreover, the majority believed that presence of bacteria in the human body can be beneficial. Participants in this study correctly identified factors that can influence microbiota such as exercise, ethnicity, or geographical location, which indicated they were aware that microbiota varies between individuals, and that certain behaviors and environmental factors can influence microbiota. Knowing the effects of certain behaviors on shaping microbiota can work as the selling point for healthcare providers to promote healthy behaviors such as exercise, and to emphasize the role of a healthy diet in shaping the gut microbiota. 

Interestingly, knowledge of microbiota was unexpectedly good in participants who reported basic/poor knowledge in microbiology. This knowledge of microbiota among university students could have been obtained through formal learning in school or university, social media, probiotic advertisings, or popular science, which is an interpretation of science intended for a general audience [21,22]. The willingness of most participants (92.8%) to learn more about microbiota and how they could affect health, should alert scientists and physicians to the importance of providing knowledge and counseling on the subject. Dispensing knowledge about microbiota and their role in health and disease can also be done through various platforms. For example, 49.1% of participants chose social media as a favorite source of information. 

One important aspect of microbiota knowledge is how antibiotics can affect microbiota, especially considering the various diseases associated with prolonged antibiotic use, and the rising antibiotic resistance in bacterial pathogens worldwide, partly due to misuse of antibiotics [23]. This study found that most participants knew the harmful effect of antibiotics on microbiota, but a substantial number of participants thought antibiotics would kill only harmful bacteria. It can be speculated that better awareness of the detrimental effects of antibiotics on microbiota might enhance antibiotic use practices among the public, thereby lessening the dangerous global rise in antibiotic resistance. This is especially relevant to Jordanian society, where antibiotic misuse is a problem, and finding new ways to address this problem is necessary. 

Several studies indicated that the use of probiotics as an adjunct therapy to antibiotics could decrease side effects of antibiotics [13,24]. In this study, probiotic use was the subject where the highest uncertainty was found among students, since almost 204 (50.7%) either disagreed or did not know that bacteria can be given to replace beneficial bacteria after antibiotic use, although it can be argued that the statement indicated that the same beneficial bacteria are replaced with probiotics, causing this high uncertainty. In contrast to the uncertainty regarding probiotic use with antibiotic therapy, the participants’ attitude towards probiotic use was positive, since around 88% of participants were willing to ingest a pill that contain bacteria to treat disease. Previous studies on knowledge of probiotics in Jordan indicated that healthcare providers have limited knowledge but positive attitudes towards probiotics [25] in line with the data in this study. 

The majority of participants in this study believed that the knowledge of microbiota could affect their lifestyle, diet, and antibiotic use. The large percentage could be due to self-selection bias and does not necessarily reflect the actual impact of microbiota knowledge on behavior. For example, no significant difference was found between different knowledge groups in the willingness to take probiotics. 

Some limitations and strengths in this study need to be addressed. First, it should be noted that the questionnaire was distributed to students through various student groups and communication channels online; thus, a true response rate could not be calculated, and a sampling bias could have been introduced, since students who were interested and possibly knowledgeable in the subject of microbiology were more likely to answer the questionnaire. Second, students had to sign in with their university email to complete the questionnaire; the university email consists of the student initials and a student identification number, which could identify the participant through the university database, but the research team was careful in removing any identifiers when processing the data, thus maintaining anonymity. On the other hand, this study could be a stepping stone for further work that assesses the understanding of important microbiological concepts among the general public, which could affect a variety of behaviors, from antibiotic use as demonstrated in this work, to hygienic practices that are important in infection control. 

This study is the first to assess knowledge of microbiota among university students outside the healthcare field and to evaluate the effect of this knowledge on behavioral beliefs. It showed that students had good knowledge of microbiota and believed that certain behaviors could be influenced by their knowledge of microbiota. With the recent advances that show the profound effect of microbiota on health, there is a need to simplify and transfer this knowledge to the public. Awareness of what makes for a “healthy” microbiota could be incorporated with other methods to encourage a healthy diet and responsible antibiotic use. Further studies are needed to find the gaps in the public understanding of microbiota and whether filling those gaps could promote healthier behaviors. 

## Figures and Tables

**Table 1 ijerph-18-13324-t001:** Demographics and general characteristics of participants.

	Count	Percent
Gender		
Male	155	38.6%
Female	247	61.4%
Total	402	100.0%
Year of study		
First year	97	24.1%
Second year	206	51.2%
Third year	73	18.2%
Fourth year	18	4.5%
Fifth year	4	1.0%
Sixth year	4	1.0%
Total	402	100.0%
Field of study		
Healthcare-related	284	70.6%
Not healthcare-related	118	29.4%
Total	402	100.0%
Reported knowledge level		
Advanced knowledge	157	39.1%
Basic knowledge	202	50.2%
Poor knowledge	43	10.7%
Total	402	100.0%

**Table 2 ijerph-18-13324-t002:** Knowledge of microbiota among participants.

Statement	Agree ^a^	Uncertain	Disagree
(1) The majority of bacteria in the world do not cause disease to humans.	324	12	66
	80.6%	3.0%	16.4%
(2) Bacterial cells outnumber human cells in our bodies.	203	68	131
	50.5%	16.9%	32.6%
(3) Presence of bacteria on the skin will always cause disease in humans.	58	9	335
	14.4%	2.2%	83.3%
(4) Presence of bacteria in the gut will always cause disease in humans	69	12	321
	17.2%	3.0%	79.9%
(5) Presence of bacteria in the brain will always cause disease in humans	277	71	54
	68.9%	17.7%	13.4%
(6) Exercise can positively affect the beneficial bacteria in the human body	320	61	21
	79.6%	15.2%	5.2%
(7) Bacteria living in the human body changes between countries and ethnicities	266	38	98
	66.2%	9.5%	24.4%
(8) Presence of bacteria on the skin can be beneficial to humans.	356	14	32
	88.6%	3.5%	8.0%
(9) Presence of bacteria in the gut can be beneficial to humans	358	17	27
	89.1%	4.2%	6.7%
(10) Presence of bacteria in the brain can be beneficial to humans	71	99	232
	17.7%	24.6%	57.7%
(11) Healthy food should never contain any type of bacteria.	80	37	285
	19.9%	9.2%	70.9%
(12) Antibiotics only kill harmful bacteria.	82	20	300
	20.4%	5.0%	74.6%
(13) Antibiotics can kill beneficial bacteria.	367	17	18
	91.3%	4.2%	4.5%
(14) Antibiotic use may cause disease by killing beneficial bacteria.	342	33	27
	85.1%	8.2%	6.7%
(15) Bacteria can be given orally to replace beneficial bacteria killed after antibiotic therapy.	198	108	96
	49.3%	26.9%	23.9%

^a^ All values are presented as count and percent.

**Table 3 ijerph-18-13324-t003:** Knowledge scores among different groups.

Variable		Knowledge Score (0–15) (Mean ± SD)	*p*-Value
Field of study	Healthcare related	11.8 ± 2.4	
	Not healthcare-related	9.6 ± 2.6	<0.001
Gender	Male	11.3 ± 2.7	
	Female	11.1 ± 2.6	0.438
Year of study ^a^	First	10.0	
	Second	11.6	
	Third	11.3	-
	Fourth	11.1	
	Fifth	13.0	
	Sixth	13.8	
Reported knowledge level	Advanced knowledge (took at least one microbiology course)	12.8 ± 2.0	
	Basic knowledge (personal reading or biology classes)	10.2	<0.001
	Poor knowledge (never read or taken a course that discusses bacteria)	9.5	

^a^ No further analysis was needed for the year of study section, since the groups are highly heterogeneous and higher years tend to contain medical students only.

**Table 4 ijerph-18-13324-t004:** Knowledge of the relationship between antibiotics and microbiota according to knowledge group.

		Advanced Knowledge (N = 157)		Basic/Poor Knowledge (N = 245)		*p*-Value	OR (95% CI)
Statement		Count	Percentage	Count	Percentage		
Antibiotics only kill harmful bacteria	Agree	13	8.3%	89	36.3%	<0.001	0.158 (0.085–0.295)
Antibiotics can kill beneficial bacteria	Agree	154	98.1%	213	86.9%	<0.001	0.130 (0.039–0.431)
Antibiotic use may cause disease by killing beneficial bacteria	Agree	152	96.8%	190	77.6%	<0.001	0.114 (0.044–0.291)
Bacteria can be given orally to replace beneficial bacteria killed after antibiotic therapy	Agree	98	62.4%	100	40.8%	<0.001	0.415 (0.275–0.626)

**Table 5 ijerph-18-13324-t005:** Effect of microbiota knowledge on probiotic use and behavioral beliefs.

		Advanced Knowledge (N = 157)		Basic/Poor Knowledge (N = 245)		*p*-Value	OR (95% CI)
Statement		Count	Percentage	Count	Percentage		
I would ingest a pill that contains bacteria as a treatment for disease if available	Agree	132	84.1%	222	90.6%	0.058	1.828 (0.997–3.351)
My lifestyle choices are affected by my knowledge of beneficial bacteria in the human body	Agree	130	82.8%	189	77.1%	0.207	0.701 (0.421–1.168)
My Diet is affected by my knowledge of beneficial bacteria in the human body	Agree	108	68.8%	162	66.1%	0.588	0.886 (0.577–1.360)
My use of antibiotics is affected by my knowledge of beneficial bacteria in the human body	Agree	152	96.8%	210	85.7%	<0.001	0.197 (0.076–0.515)

**Table 6 ijerph-18-13324-t006:** Willingness to learn about microbiota.

.		Advanced Knowledge (N = 157)		Basic/Poor Knowledge (N = 245)	
Statement		Count	Percentage	Count	Percentage
I would like to learn more about how bacteria in the human body can affect health and disease	Yes	150	95.5%	225	91.8%
	No	7	4.5%	20	8.2%
If yes, which of the following sources would you use? (You can choose multiple options)	Trusted medical sources	112	71.3%	172	70.2%
	Healthcare workers	53	33.8%	96	39.2%
	News outlets	19	12.1%	42	17.1%
	Social media	69	43.9%	114	46.5%

## Data Availability

All data generated or analyzed during this study are included in this published article.

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
