# Peer review of "Evaluating Knowledge of Human Microbiota among University Students in Jordan, an Online Cross-Sectional Survey"

_ijerph, 2021, doi:10.3390/ijerph182413324_

Round 1
Reviewer 1 Report
Dr. Anas Abu-Humaidan and colleagues present the results of a survey of knowledge on microbiota and human health that was conducted among medical and non-medical students. The survey was brief but highlights a few interesting trends, especially with regard to beliefs on antibiotics.
- Abstract: please consider including information on the duration of the microbiology course (in brackets), e.g. … a formal Microbiology course (40 hours) or not.
- Abstract: “Behavioral beliefs of participants regarding antibiotic use, but not diet or 23
- probiotic use, were significantly .” - sentence left unfinished
- No reason to capitalize the word “Microbiology” in the abstract
- I imagine it was difficult to create adequate questions. The use of terms “always” or “can” in multiple choice question often suggests the answer. This, however, cannot be changed now that 400 persons have answered the surveys. It might have been more difficult for the students to agree or disagree with general statements “The presence of bacteria in the brain causes disease” – however, it would have also been more difficult to categorize the answers as correct or incorrect and interpret the answers.
- “(9%) of participants disagreed that bacteria can be given orally to re- 160
place beneficial bacteria killed after antibiotic therapy” – I could disagree also, because the probiotics may: (1) not replace exactly the same commensal bacteria that have been eradicated by an antibiotic, (2) inhibit the return of species’ proportions to the state prior to antimicrobial therapy. Some probiotic species have a proven important role in the prevention of antibiotic-associated diarrhea, but this does not prove they restore healthy microbiota. Even with the use of probiotics, repeated antibiotic therapies will continue to reduce the diversity of microbiota encountered within our societies. Therefore – I think that both the authors of the question and the responded who disagreed are correct, in different aspects. “indicating the highest level of uncertainty among all statements” – therefore, this is unsurprising! It is indeed an interesting and important question. - In many countries, health-related fields tend to attract a greater percentage of female students. Was the percentage of women greater in healthcare-related schools?
- It might have been expected that medical students have better knowledge of microbiota than other students. However, it is interesting to see that this gap is mostly related to the interpretation of the effects that antibiotics may have. For healthcare professionals it may therefore be an important point to educate the society on this specific point. A new, brief survey could be conducted among nurses and doctors, and patients of general practice offices, to confirm that this trend also exists within the society, and probably is even more pronounced than in students. The concept that antibiotic only kills harmful bacteria may strongly impact patient expectations towards the healthcare system, leading to more use of antibiotics.
- Line 200 “Moreover, higher knowledge also agreed more” I suggest “students with better knowledge also agreed more…”
- A smaller percentage of respondents agreed with the fact that the knowledge of microbiology influences dietary choices. This would suggest that the course in microbiology should include content on the susceptibility of microbiota to dietary-induced change and examples of such beneficial dietary interventions.
- It is not clear for me if advanced knowledge was self-declared, self-declared according to criteria (explanation to respondent that advanced knowledge means that a microbiology course was taken) or even somehow verified formally.
- The percentage of students interested in learning more about microbiota is very high and probably reflects the understanding of the importance of this topic. However, it should be made clear what percentage of the invited students took the survey (I do not see this in the results). Those who answered the invitation may have been interested in microbiota from the start (a source of bias? – should it be very briefly discussed?).
- Line 248 “Furthermore, we found that antibiotic Better awareness” please correct
- Lines 248-250 not exactly this was not found by the study but it is a logical and valid hypothesis – it can be speculated that …
- Line 256 “different” or “diverse”?
- Line 276 please consider: “for example” not necessary (“or” would need to be replaced by “and” or modified otherwise), or change the sentence structure by moving “for example”.
- The conclusion is adequate and supported by study findings.
Reviewer 2 Report
The study presented is relevant and based in recent publications and analyses.
Line 12, 19, 24, etc - "we aimed"; "Our results"; "Our study" - Science is written in the third person, and should not be personalized with the use of "we", "our", etc. Please, correct all the text regarding this issue.
Line 196 - "p value > 0.05)." - Verify please.
A better explanation about the rationale related to relevance of this study and for the Jordanian society. The impact of this study to the general population should be better addressed and discussed as well.
Reviewer 3 Report
The aim of this study was to investigate the knowledge of university students on human microbiota and the impact on their behavioral beliefs. To this aim an online questionnaire was distributed among students studying various scientific fields, both health-related and non-health related. Compared to students with basic/poor knowledge of microbiota, significantly higher proportions of participants with advanced knowledge believed that antibiotics can kill not only harmful bacteria but also the beneficial ones and cause microbiota – related diseases. However, this knowledge does not seem to affect their behavioral beliefs on diet and lifestyle.
The study is relative with the content of the section “Health Behavior, Chronic Disease and Health Promotion”. However, the manuscript needing extensive revision. In short, description of statistical analysis is incomplete, while presentation of the results needs revision. There are some errors in data analysis regarding the “ p ” and “ % ”. The tables need extensive revision. The statistical significance of certain between-groups comparisons is not presented either in the text or the tables, making the precise interpretation of results impossible. The strengths and limitations of the study are not presented. The conclusions are not totally supported by the results. English need extensive revision. The references do no follow the journal’s style.
Major comments
ABSTRACT
- Lines 23-3. “Behavioral beliefs of participants regarding antibiotic use, but not diet or probiotic use, were significantly .” This sentence is not understandable
INTRODUCTION
2. Lines 57-8. “secondly to assess the effect of this knowledge on the attitudes and lifestyle choices of students”
Comment: The items included in the questionnaire (as presented in the main text) do not adequately assess the effect of knowledge of microbiota on the student’s attitudes and life style. Which aspects of life style were assessed? How did the knowledge affect the diet??
METHODS
3. Lines 67-7. “For verification purposes, the participants needed to sign in with their university email.”
Comment: Given that the students signed with their email, the survey was not anonymous. This fact is an important source of bias and must be commented in the limitation section.
4. Lines 71-72. “The questionnaire can be found in the supplementary material.”
Comment: The supplementary material is not described at the end of the main text. Moreover, I could not find any supplementary material among the submitted files.
5. Lines 82-3. ” the level of knowledge about bacteria estimated by the participant’s enrolment in specific Microbiology courses.”
Comment: To my opinion, grouping the responders into those with advanced and basic/poor knowledge of microbiota based only on the attendance of microbiology courses does not precisely assesses the level of knowledge. Moreover, it is possible that the great majority of the students with advanced knowledge study health sciences. This issue must be clarified in the result and discussion sections.
6. Line 90. ” The statements used estimated the participant’s knowledge about microbiota and how it can affect health.”
Comment: The questionnaire could estimate the participants’ knowledge on whether the microbiota can affect heath or not, but not how it can affect it, since no question addressed the aspects of health that could potentially be affected by the microbiome.
7. Data analysis
Line 109. “Data are presented as count and percentages for categorical variables.”
Comment: The presentation of continuous variables, i.e. the score of microbiota knowledge, (mean, median, SD, IQR??) is not mentioned either in the data analysis section, or in the result section, or in the Tables.
RESULTS
8. General comments: (a) the results must be presented in past tense in the result section. (b) the first letter of the word “table(s)” must be capital throughout the text. (c) Most counts and percentages are presented incorrectly as regards their position in/out of the parenthesis, as shown in the following example:
9. Lines 115-22. “Among them (155, 38.6%) were males, whilst (247, 61.4%) were females, participants were distributed in the age groups 17-20 years (350, 87.1%), 21-24 years (51, 12.7%) and one response of the age 29 (0.2%) (table 1.). Respondents were grouped by their field of study as health-related specialties (284, 70.6%) and non-health related (118, 29.4%). The year of study with the greatest prevalence was the second year (206, 51.2%). As for the self-reported knowledge level (described in methods), (157, 39.1%) of the sample reported they have advanced knowledge about Microbiology, while (202, 50.2%) reported they had basic knowledge, and (43, 10.7%) had poor knowledge.”
The correct presentation is as follows: “Among them 155 ( 38.6%) were males, whilst 247 (61.4%) were females. Participants were distributed in the age groups 17-20 years (n= 350, 87.1%), 21-24 years (n= 51, 12.7%) and one response of the age 29 (0.2%) (Table 1.) Respondents were grouped by their field of study as health-related specialties (n=284, 70.6%) and non-health related (n=118, 29.4%). The year of study with the greatest prevalence was the second year (n=206, 51.2%). As for the self-reported knowledge level (described in methods), 157 (39.1%) of the sample reported they have advanced knowledge about Microbiology, while 202 (50.2%) reported they had basic knowledge, and 43 (10.7%) had poor knowledge.”
This style must be kept throughout the text.
10. Line 130. “(80.6%,50.5%,…)» add an and «(80.6% and 50.5%,…)»
11. Lines 145-6. “We also evaluated if participants were aware of the dynamic nature of microbiota and how it is affected by environmental and genetic factors, …”
Comment: To my opinion, the questions of this survey cannot sufficiently support this statement.
12. Lines 186-7. “Similarly, (91.7%) of advanced knowledge students agreed to the statement “antibiotics only kill harmful bacteria” vs. (63.7%) of basic/poor knowledge students (Table 4).”
Comment: This statement is wrong. In fact, the opposite is correct.
13. Line 198-200. “My lifestyle choices are affected by my knowledge of beneficial bacteria in the human body” (82.2 % vs 77.1%, respectively, p value < 0.001).”
Comment: The p is incorrect. The right p is 0.207, which changes the interpretation of this result.
14. Lines 209-12. “Majority of participants (92.8%) wanted to learn more about microbiota (table 6). The top two sources chosen were social media (49.1%) and trusted sources (76.1%).”
Comment: The percentages are wrong. The correct figures are as follows: “Majority of participants (93,2%) wanted to learn more about microbiota (table 6). The top two sources chosen were social media (45,5%) and trusted sources (70,6%).”
15. Tables: General comments:
(a) The statistical significance (p) should be added in the Tables showing comparison between groups.
(b) Usually the “p” is presented with 3 decimal places, either significant or not.
(c) When the answer choices are two (i.e. yes/no, agree/disagree) only the positive responses (counts and %) can be presented in the tables along with the p (and possibly the Odds and CI, when applicable), since the respective “no” or “disagree” is self-evident.
Suggested table format
|
|
|
Advanced knowl (N=157) |
Basic/poor knowl (N=245) |
|
|
|
|
|
Count (%) |
Count (%) |
p |
OR (5 - 95 CI) |
|
Antibiotics only kill harmful bacteria |
Agree |
13 (8.3) |
89 (36.3) |
…. |
…. |
|
Antibiotics can kill beneficial bacteria |
Agree |
154 (98.1) |
213 (86.9) |
…. |
…. |
DISCUSSION
16. Lines 228-31. “Our study indicated that university students did not necessarily associate bacteria with disease, in fact, most participants agreed that bacteria can be found in different loca tions of the human body without causing disease, and in some instances, presence of bac teria in the human body can be beneficial”.
Comment: Please rephrase
17. Lines 231-2. “Participants in our study were also aware of the dynamic nature of microbiota, and correctly identified factors that can influence microbiota such as exercise and genetic diversity.”
Comment: How the “genetic diversity” was identified? Which results support this statement?
18. Lines 234-6. Please rephrase
19. Lines 237-8. What is the definition of “popular science”?
20. Lines 265-8. “Nevertheless, it indicates that when deciding on matters such as diet or antibiotic use, evaluating the impact of those behaviors on microbiota is not farfetched, even in those with basic understanding of Microbiology.”
Comment: I don’t understand the meaning of this statement. Which results support it? How this sentence is connected with the previous one?
21. Line 271-2. “It showed that students have good knowledge of microbiota and are willing to consider the impact of certain behaviors on microbiota.”
Comment: Which results support this statement? Of note, the correct p shows that the percentage of positive responses to the questions regarding the effect of knowledge level on life style and diet did not differ significantly between participants with good knowledge of microbiota and those with poor knowledge. Moreover, this conclusion (if this is the conclusion of the study) differs from the conclusion in the abstract “Our study indicates that disseminating knowledge regarding mi crobiota and Microbiology in general, can improve behaviors related to antibiotic use.”
22. Comments on the strength and limitations of the study are missing
REFRENCES
23. The references do no follow the journal’s style
24. Minor comments:
Change “vs” to “vs.” throughout the text.
Line 270. “the effect of this knowledge behavioral beliefs”. Please, change to “the effect of this knowledge on behavioral beliefs”
Round 2
Reviewer 3 Report
Reviewer’s comments on the revised manuscript
Dear authors,
You have adequately addressed all my comments on the first version of the manuscript and the paper has been considerably improved. However, some minor English revision is still needed. Following there are some examples of the points needing English corrections (grammar errors).
Lines 15-6 (abstract): “Using an online questionnaire that consisted of 3 parts: 15 demographics, general knowledge of microbiota, and behavioral beliefs related to microbiota.” This sentence is not complete (the verb is missing)
Line 98. “…… cross-sectional study, the sample consisted of students ….” The comma must be replaced by a full stop
Line 105. “……. University of Jordan; they needed …..” The semi-colon must be replaced by a comma
Lines 106-7 “…… student ID, the student ID cannot be ….”.Please, change to “…… and their student ID, which cannot be ….”
Line 116. “into three sections; 1) Demographics and …” The semi-colon must be replaced by a colon (:).
Lines 127-30. “The answers (probably and definitely) are referred to as (agree) in the results section and the answers (definitely not and probably not) are referred to as (disagree), while the answer (I don’t know) is referred to as (uncertain).” All parenthesis must be replaced by quotation marks (“..”).
Lines 161-3. Similar changes as above are needed
Lines 297-8. “…. was measured, the majority (80.6% ..................... above-mentioned statements (Table 2), this indicated ……”. Both commas must be replaced by full stops.
Lines 305-6. “…. is not considered to harbor microbiota, majority of participants thought correctly ….”. Change to “…. is not considered to harbor microbiota. The majority of participants thought correctly ….”
Lines 313-5. The sentence must be rephrased as it is grammatically incorrect.
Line 345. “Participants knowledge on the relationship …” change to “Participants’ knowledge on the relationship….”
Line 362. “with a score of (10.1 ± 2.5). The…”. The parenthesis must be deleted.
Line 365. “ … study was done, students in ..”. The comma must be replaced by a full stop.
Line 369. “p<0.001). While no significant ..”. The full stop must be replaced by a comma.
Lines 533-5. “… other students, since (98.1% vs. 86.9%, respectively, p < 0.001) agreed that antibiotics can kill beneficial bacteria, and (96.8% vs. 77.6%, respectively, p < 0.001) agreed that antibiotic use..”. Please change to ““… other students, since 98.1% vs. 86.9%, respectively (p < 0.001) agreed that antibiotics can kill beneficial bacteria, and 96.8% vs. 77.6%, respectively (p < 0.001) agreed that antibiotic use..””.
Lines 536-7. “.. kill harmful bacteria vs. (36.3%) of basic/poor knowledge ..”. Parenthesis must be deleted.
Line 636. “However, students with better knowledge agreed more with the statement …”. Please change to “However, more students with better knowledge agreed with the statement..”
Lines 797-8. “ This proved challenging, for example, we had to …”. Please rephrase.
Line 811. “… not always associated with disease, moreover, the majority believed ....” Please change to “..not always associated with disease. Moreover, the majority believed ..”
Line 821. “…. could have been obtained through: formal learning in school or…..”. The colon must be deleted.
Line 827. “various platforms, for example…”. Rephrase to “… various platforms. For example..”
Line 841. “.. since almost (204, 50.7%) either disagreed …”. Change to “since almost 204 (50.7%) either disagreed…”.
Line 1051. “ .. since around (88%) of participants ..”. Parenthesis must be deleted.
Line 1058. “… on behavior, for example, no …”.Change to “… on behavior. For example, no …”
Line 1069. “…. data, thus anonymity was maintained.” Please change to “..data, thus maintaining anonymity.”
